# Abundant Citizen Science Data Reveal That the Peacock Butterfly *Aglais io* Recently Became Bivoltine in Belgium

**DOI:** 10.3390/insects12080683

**Published:** 2021-07-29

**Authors:** Marc Herremans, Karin Gielen, Jos Van Kerckhoven, Pieter Vanormelingen, Wim Veraghtert, Kristijn R.R. Swinnen, Dirk Maes

**Affiliations:** 1Natuurpunt Studie, Coxiestraat 11, B-2800 Mechelen, Belgium; karin.gielen@natuurpunt.be (K.G.); studie@natuurpunt.be (J.V.K.); pieter.vanormelingen@natuurpunt.be (P.V.); wim.veraghtert@natuurpunt.be (W.V.); kristijn.swinnen@natuurpunt.be (K.R.R.S.); 2Research Institute for Nature and Forest (INBO), Herman Teirlinckgebouw, Havenlaan 88 Box 73, B-1000 Brussels, Belgium; dirk.maes@inbo.be

**Keywords:** peacock butterfly, *Aglais io*, life history strategy, citizen science, change in voltinism, bivoltine, Belgium, climate change

## Abstract

**Simple Summary:**

The peacock butterfly is abundant and widespread in Europe. It used to have a single generation per year: adults born in summer overwintered and reappeared in spring to reproduce. However, recent flight patterns in western Europe show three peaks during the year: a first one in spring (overwintering butterflies), a second one in early summer (offspring of the spring generation), and a third one in autumn. Hitherto, it was unclear whether this third autumn flight peak was a second new generation or consisted of butterflies flying again in autumn after a summer rest. Based on hundreds of thousands of observations and thousands of pictures submitted by naturalists from the public to the online portal ‘observation’ in Belgium, we demonstrate that Peacocks shifted towards two new generations per year in recent decades. Mass citizen science data has become increasingly important in tracking the response of biodiversity to rapid environmental changes (e.g., climate change).

**Abstract:**

The peacock butterfly is abundant and widespread in Europe. It is generally believed to be univoltine (one generation per year): adults born in summer overwinter and reappear again in spring to reproduce. However, recent flight patterns in western Europe mostly show three peaks during the year: a first one in spring (overwintering butterflies), a second one in early summer (offspring of the spring generation), and a third one in autumn. It was thus far unclear whether this autumn flight peak was a second new generation or consisted of butterflies flying again in autumn after a summer rest (aestivation). The life cycle of one of Europe’s most common butterflies is therefore still surprisingly inadequately understood. We used hundreds of thousands of observations and thousands of pictures submitted by naturalists from the public to the online portal observation.orgin Belgium and analyzed relations between flight patterns, condition (wear), reproductive cycles, peak abundances, and phenology to clarify the current life history. We demonstrate that peacocks have shifted towards two new generations per year in recent decades. Mass citizen science data in online portals has become increasingly important in tracking the response of biodiversity to rapid environmental changes such as climate change.

## 1. Introduction

Many aspects of the life of ectotherms such as biochemistry and development speed depend on environmental temperatures [1,2,3]. Climate change may therefore fundamentally affect their life history, and changes in their phenology are most notable [4,5,6,7,8,9,10,11,12]. There is ample evidence that climate warming also has fundamental, yet sometimes complex, impacts on the life history of Lepidoptera [13,14,15,16,17,18]. As development speeds up and the time window of optimal temperatures widens, phenological shifts can sometimes even allow sufficient time to fit in an extra generation (change in voltinism) [19,20,21,22,23,24,25,26,27,28,29,30,31,32].

Our study species is the peacock butterfly, a widespread and abundant species over most of Europe [33,34,35,36]. A single generation is the norm [37,38,39,40,41,42,43,44]: overwintering butterflies fly and reproduce in early spring and the new generation appears in early summer and goes into diapause from mid-summer onwards. A phenogram for its entire range indeed shows in essence two flight peaks [45], compatible with one overwintering generation. However, its life history seems to vary regionally: generalized statements in textbooks indicate that it has a single generation, but that a (partial) second or even a third generation may also occur, at least occasionally [33,34,35,39,40,41]. However, in continental Europe, we found limited recent phenological data with only two neat flight peaks supporting one generation (univoltinism). Recent phenograms for the European continent show a small [37,44,46,47,48,49] to fully developed third flight peak [49,50,51,52,53] in late summer or autumn, six to ten weeks after the early summer peak. The possibility of a second generation is frequently mentioned [37,39,44,46,47,49,50,51,52,54,55,56,57,58,59,60], but only a minority of authors interpret the data as a bivoltine life cycle [39,49,50,51,52,56,57,58,59,60,61]. The extent of the second generation is said to be variable, dependent on region, year, or weather conditions [33,34,37,39,41,46,47,50,52]. Whether this third flight peak in autumn represents a genuine second generation is obscured by the possibility that peacocks of the second flight peak may soon after hatching go into diapause early in summer (i.e., aestivation), and may reappear later in autumn to top up fat stores before winter, causing two flight peaks in one year from the same butterflies [37,52]. The life cycle of one of Europe’s most common butterflies is therefore surprisingly inadequately understood.

Here, we assessed the current life cycle of the peacock butterfly in Belgium from abundant citizen science data. We analyzed hundreds of thousands of observations and pictures submitted to the online wildlife data portal https://observation.org (accessed 1 February 2021). We investigated seasonal patterns of abundance of butterflies and classified the state of wear of butterflies in pictures to distinguish between new waves of emergences from pupae as opposed to reappearances in autumn after a short summer diapause. The timing of reproductive stages (eggs, caterpillars, pupae) was also assessed. We correlate the abundance of each generation with the abundance of next year’s spring peak to check which generation contributes most to the next year. This way, we demonstrate that the peacock recently had two generations in Belgium, one in early summer and one in late summer, the latter one reappearing the following spring: it is therefore now bivoltine rather than having a single generation. From data since 1950, we show that this change is recent and fast. The large volume of citizen science data and pictures proved vital to unravelling the current life cycle of one of our most common butterflies. 

## 2. Materials and Methods 

We analyzed occurrence records and pictures of peacock butterflies submitted by citizen scientists to the online biodiversity observation portals https://waarnemingen.be (in Dutch) (accessed 1 February 2021) and https://observations.be (in French) (accessed 1 February 2021), the local versions in Belgium of the international platform https://observation.org (accessed 1 February 2021). With over 47 million recent records of >23,000 species from 30,689 km^2^, this platform generated one of the most dense datasets of its kind in the world. We used the verified data from 12 years (2009–2020): records of 319,684 adult peacock butterflies, 77 records of eggs, 2894 of caterpillars, and 60 of pupae. 

For the reconstruction of flight peaks, an abundance index was calculated by relating records to search efforts based on the evidence of field activities left in the database by proficient observers. For this search-effort correction, we selected observers with at least 50 records of at least 10 butterfly species each year. As a measure of their collective search effort, we calculated the sum for each day of all the 100 × 100 m grid cells from which an observer had reported records (of any species, including non-butterflies; also including absence records). This method of ‘proven day-grid-visits’ has become the standard proxy for search effort when analyzing incidental observations of the portal waarnemingen.be [62,63,64,65,66]. Day-grid-visits do not cover search effort completely, because records are not submitted from every visited hectare grid cell, but strongly correlates with it. The selection of proficient observers resulted in a subset of 1525 observers who reported in total 250,651 peacock butterflies during 6,741,517 hectare-day-grid-visits.

For each day we calculated the index of abundance as the quotient of the total number of butterflies reported relative to the total number of day-grid-visits. The abundance index per week is not simply the sum of all seven days, because days with good or poor weather can heavily affect the number of butterflies detected, and that number of suitable days may differ considerably per week and between years for the same week. We therefore only used the average of the three best days in each week, i.e., with the highest day-index (highest ratio of butterflies to day-grid visits). Thus, of the above selected numbers, for the weekly abundance indices we only used 178,874 peacock butterflies recorded during 3,407,664 day-grid-visits. The flight performance index for an entire generation is the sum of the performance indices over the weeks the generation flies (hence, a measure of ‘area under the curve’ in the abundance phenogram).

Like Vodičková and colleagues [67], we derived voltinism from wing wear. To investigate patterns of wing wear, we screened 15,859 pictures of peacock butterflies from Belgium. After discarding dead and dormant individuals and rejecting pictures of insufficient quality and of individuals with closed wings, we classified the condition of wear of the upperwings in 12,425 butterflies according to four categories: (1) immaculate: fresh, or with at most one tiny scratch on the upper wings or one minor dent to the wing edges, (2) slightly worn: some scratches on the upper wing and/or small dents to the edges, (3) moderately worn: many scratches to the upper wing and/or dented edges, and (4) heavily worn: (parts with) colors faded and/or heavily dented edges. The categories were chosen to allow the distinction of recently emerged individuals in particular: there is “more wear” between categories 3 and 4 than between the first two. Persistent wear and abrasions with multiple impacts at several spots was given more weight in the classification than a single major incident that resulted in larger parts of the wing missing. Pictures were randomly sorted before being classified by a volunteer (JVK), who worked from a series of reference examples (e.g., Figure 1, see more details and examples in Appendix A) and who was unaware of the research hypothesis when scoring the pictures.

Pictures submitted by the public are not necessarily representative. As this could compromise the patterns of wing wear deduced from pictures, we investigated the possibility that people are more keen on photographing fresh and good-looking butterflies rather than worn ones. Fortunately, observers showed no particular preference, and pictures of peacocks submitted by the public to the online portal are a reasonable mix of the good, the bad, and the ugly, without any bias for the nice (Appendix B: Figure A2, Figure A3, Figure A4, Figure A5, Figure A6 and Figure A7).

Finally, in regions and years with a second generation, the question arises which of the two generations contributes (most) to the overwintering population. Do all the butterflies of the first generation lay eggs and contribute to the second generation in autumn? Or is there still a mixed strategy with only a partial second generation, meaning that substantial numbers of butterflies from the first generation in early summer go straight into dormancy without laying eggs, only to re-emerge the following spring? We investigated this by correlating the abundance of each generation of a certain year with the abundance of the next spring peak. Under the first hypothesis, the autumn peak should show a strong correlation. Under the second hypothesis, the early summer peak or the combined abundance of both peaks should show a strong correlation. We used Past3 online software [68] to calculate a multiple linear model with one dependent variable (next spring abundance) and the abundance of each generation as two independent variables.

To investigate historical changes, we calculate for each year with sufficient data the proportional abundance of the third flight peak since 1950 for Flanders (northern part of Belgium) (based on the data used by [69]). 

## 3. Results

### 3.1. Flight Peaks

During the most recent decade, the phenology of peacock butterflies shows three distinct peaks: the first peak comes from overwintering adult peacock butterflies becoming active again in spring (March–May), a second flight peak in early summer (July) and a third peak in late summer and early autumn (September–October) (Figure 2). The spring peak is the smallest, the late summer peak the largest: the second flight peak is c. 83% larger than the first and the third another 23% larger than the second (comparing area under the curve in Figure 2). Notwithstanding substantial annual variation of the abundance in each flight peak, there was a large third flight peak in each of the 12 recent years (its size varying between 35% and 543% (on average 154%) of that of the second peak) (see Appendix C: Figure A8a–l). 

### 3.2. New Generations and Wing Wear

Peacock butterflies re-appear in spring in a wide range of wing conditions: some still have immaculate wings after the winter diapause, others emerge already heavily worn (Figure 3 and Figure 4). During the reproduction period in spring, the wear rapidly increases until nearly all of the few remaining butterflies are heavily worn by late May or early June. A first new generation of immaculate butterflies emerges in late June, early July, when up to 80% are in immaculate condition. While abundance declines again strongly in August (Figure 2 and Figure 3), the wear increases more slowly than in spring (Figure 4). Some 8–10 weeks after the summer generation, a new generation of immaculate butterflies emerges over several weeks from late August to early October (Figure 3 and Figure 4). They may fly until November, but the wear increases even more slowly compared to spring or summer. 

### 3.3. Reproduction

Eggs of and egg-laying by peacock butterflies were observed from April to mid-May and again from late June to mid-August (Figure 5). There are also a few records of eggs around mid-September, indicating some attempts at a third reproductive cycle. Caterpillars were mainly recorded in two clear peaks, the first from late May to early July and the second in August and early September (Figure 5). There were on average 10% fewer records of reproduction in the second generation than in the first, and they occur less concentrated in terms of time. 

There are still 33 records of caterpillars from October onwards, a further indication of occasional attempts at a third generation. Pupae are rarely reported, mainly during the second half of June and from mid-August into October. There are, however, also a few records of pupae during November and December, and even in spring (March–April), but there is no proof that these were still viable. On average, there were 71 days between the median dates of both caterpillar peaks. Inter annual variation in timing of reproduction fluctuated up to four weeks for the first and five weeks for the second reproductive cycle. In particular, the second period of reproduction has shifted to earlier dates in recent years (Figure 6). 

### 3.4. Contribution to the Next Years’ Spring Peak

We investigated the correlation between the abundance of the spring generation (dependent variable) and the abundance of either of the two generations the year before (independent variables) with a multiple linear regression model: the contribution of the second generation to the abundance in spring next year is highly significant (*R*^2^ = 0.80, *P* < 0.0002), while the contribution of the first generation is not significant (*R*^2^ = 0.17, *P* = 0.21) (Figure A8a). We may therefore conclude that it is the second generation that constitutes the bulk of the overwintering population with little contribution from the first generation of peacocks flying earlier in summer. Peacocks therefore now fly in a true bivoltine pattern in Belgium. Two extreme years convincingly illustrate the contribution of the second generation: (a) the exceptionally abundant second generation in autumn 2016 was followed in 2017 by the most abundant spring population of the 12-year study period (overall peacocks remained most abundant of all years in 2017) and (b) the failed second generation in autumn 2018 (because of drought) resulted in the poorest overall numbers in 2019. The second generation therefore contributed most to the strongest annual fluctuations. In fact the coefficient of determination is much higher between the abundance of the second generation in autumn and the population emerging the following spring (*R*^2^ = 0.80, *P* < 0.0002) than between the spring population and its descendants in the summer of the same year (*R*^2^ = 0.11, *P* = 0.30) or between the first and second generation of the same year (*R*^2^ = 0.06, *P* = 0.44). This is not really a surprise since only adult mortality during winter stands between the second generation in autumn and the abundance the following spring, while generations within a year connect through reproduction, which involves several stages of the life cycle, each affected by a complex series of regulating factors and risks.

### 3.5. Fast Recent Increase of Bivoltiny

Historically, there have been some records of peacock butterflies during autumn in most years. They contribute <15% to the annual total observations (mostly even <5%). It is unclear whether these were occasional second generation imago’s or a few adults re-emerging from dormancy after summer. With three flight peaks, the third peak (second generation) can be expected to take up some 30% of the annual total. When the abundance of the second generation reaches only 15–25% of the annual total, there is rather partial bivoltinism, with part of the adults of the first generation reproducing the same year and others going into dormancy without reproducing; or the second generation may have failed (like in 2018). Once the third flight peak reaches over 25% of the annual total, we can assume that the majority is bivoltine. 

Before 1975, a second generation was rare and at most partial in the peacock butterfly in Belgium. The first evidence of a full second generation occurred in 1976, when the second generation accounted for 33% of the total annual abundance. The next bivoltine year was 1989 (reaching 43%). From 1990–2004, partial bivoltinism became more regular in Flanders (Figure 7): years with partial bivoltinism alternated with univoltine years (<5% of annual total: 1993, 1996, 1998, 2000). Since 2005, bivoltinism has become the rule, with the second generation accounting for >25% of the annual total in most years (Figure 7). Although there is a clear increasing trend over the years, the abundance of the second generation (and hence voltinism) can change dramatically between years (see, e.g., 1994–2000, 2004–2007). The peacock sometimes switched between univoltine and bivoltine from one year to the next (e.g., 2004–2005 and 2006–2007). 

## 4. Discussion

The unprecedented number of occurrence records, including pictures and immature stages, recently gathered in the online citizen science data portal https://observation.org (accessed 1 February 2021) was vital for the interpretation of the third flight peak of the peacock butterfly’s life history in Belgium. 

Six weeks is sufficient time for peacock butterflies between egg-laying and emergence of the next generation of adults [70] and therefore the two reproductive peaks documented here (Figure 5) are perfectly complementary to the three flight cycles (Figure 2 and Figure 3), strongly corroborating a life history with two local generations per year. With three flight peaks per year, usually of growing importance, appropriately spaced (Figure 2 and Figure 3), reproduction documented at the right time in between (Figure 6), large numbers of fresh butterflies suddenly emerging at the onset of the second and third flight peak (Figure 3 and Figure 4), and with the second generation constituting at least the bulk of the overwintering population (see also [50,52,61]), there is no doubt that peacock butterflies were fully bivoltine in Belgium recently. We had the advantage in this study that Belgium is a relatively flat country where the local phenological patterns are little at risk of becoming obscured by individual butterflies undertaking altitudinal movements from zones with a different phenology or voltinism. 

The slower wing wear in the population during the flight peaks in summer and autumn than in spring may indicate that emergence of new butterflies from the pupae is gradual over several weeks, providing a continuous source of new, immaculate butterflies. During the autumn peak, more worn individuals may also disappear as they go into diapause. As there are more immaculate butterflies during the third flight peak than during the second, and because in some years butterflies are substantially more numerous in the third flight peak than in the second, this is a new generation and cannot be the same butterflies re-appearing again after a summer rest, further corroborating bivoltinism.

Based on the appearance of an extra flight peak, at least partially successful attempts at additional generations have recently been found in almost half of the butterfly species in Belgium, particularly during exceptionally hot years [49,64]. A phenological shift and an extra generation may be beneficial in polyvoltine species[27], but if the generation comes too late in the season, it may also constitute a developmental trap [32]. Climate extremes in general [71], and an extra generation later in summer in particular, may make caterpillars of an additional generation more sensitive to drought [22], which causes food plants to wither. This is also mentioned for peacocks [64,72] and shown in our data from the failing second generation in autumn 2018 (Figure A8j). Despite their response to climate warming with an extra generation, peacocks are actually declining in Belgium [49]. Some other butterfly species in which the larvae feed on common nettles *Urtica dioica* are faring even worse. The small tortoiseshell *Aglais urticae* recently moved up the local Red List to endangered and is currently amongst the fastest declining butterflies in Flanders (northern Belgium) [69]; the decline is probably related to climate warming [38,39,73]. The map *Araschnia levana*, now also producing a (partial) extra generation [64], has collapsed during recent hot years [69].

Judging from a similar pattern of three flight peaks, bivoltinism has also been the case in peacock butterflies recently in the wider area of continental Europe: at least in northern France [44,51], southern Germany [46,50,74], Switzerland [56,57,58] and Slovakia [59]. Further north, e.g., in Scandinavia [75,76], but also in the UK [38,42,43,77], peacock butterflies seem to still be strictly univoltine.

The first incidence of bivoltiny in Belgium occurred in 1976, an exceptionally long, hot summer [78] and the next was in 1989, the first year with a temperature anomaly over +1.5 °C. An extended flight window caused by climate warming is involved in the change in voltinism in the peacock butterfly, but the spatiotemporal patterns and their environmental correlates and mechanisms are the subject of a separate paper.

## 5. Conclusions

While peacock butterfly was historically univoltine, abundant citizen science data on flight peaks, reproduction and state of wear showed that the species became bivoltine in Belgium recently.

Availability of abundant citizen science data including butterfly pictures and immature stages from the online naturalist data portal https://observation.org (accessed 1 February 2021) was vital for the reconstruction of the peacock butterfly life history, and more specifically the distinction between alternative hypotheses for its flight pattern.

Large scale, mass citizen science data are a crucial resource to track the impact of the current fast environmental changes on biodiversity.

## Figures and Tables

**Figure 1 insects-12-00683-f001:**
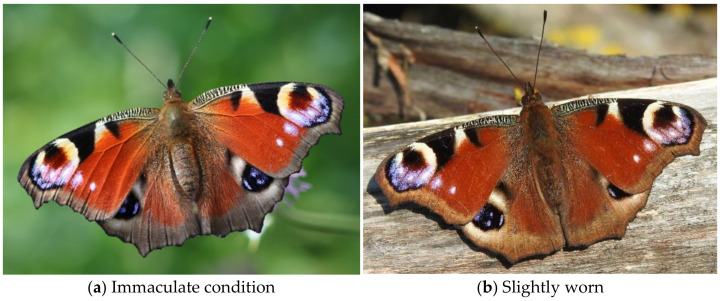
Examples of classification of the condition of wing wear in peacock butterflies.

**Figure 2 insects-12-00683-f002:**
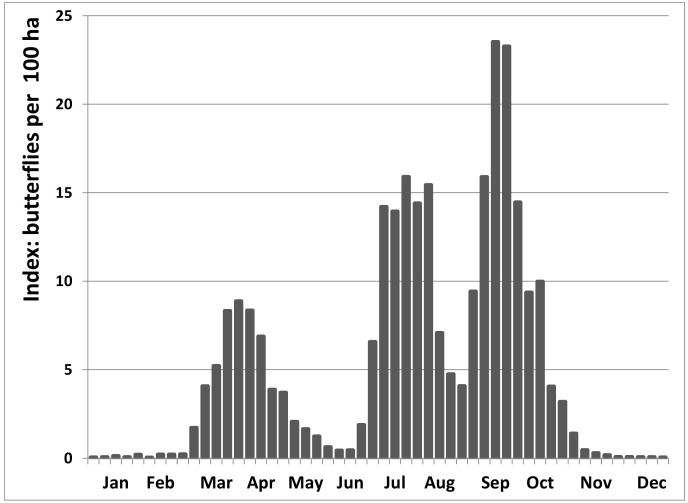
Abundance indices per week of peacock butterflies in Belgium (2009–2020): number of butterflies reported (*n* = 178,874) relative to the number of visited hectares (day-grid-visits: *n* = 3,407,664) for the three best days of the week (see methods).

**Figure 3 insects-12-00683-f003:**
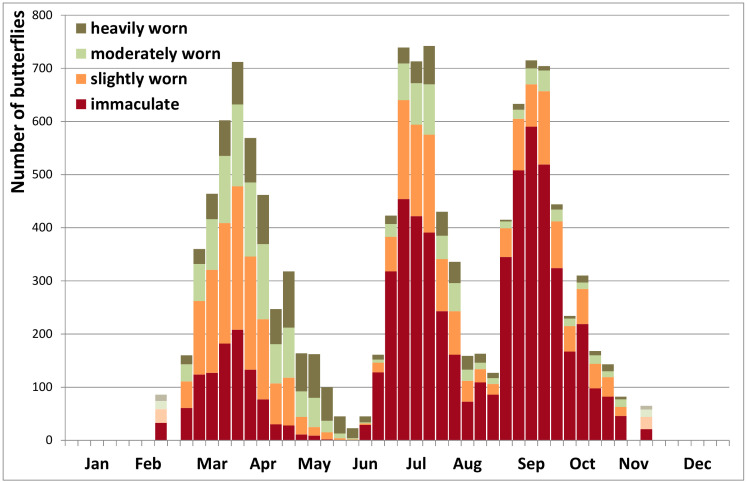
Phenology of wing wear of peacock butterflies in Belgium 2009–2020: numbers by week (*n* = 12,425). (The lower numbers before week 10 and after week 44 were lumped and plotted at weeks 8 and 46).

**Figure 4 insects-12-00683-f004:**
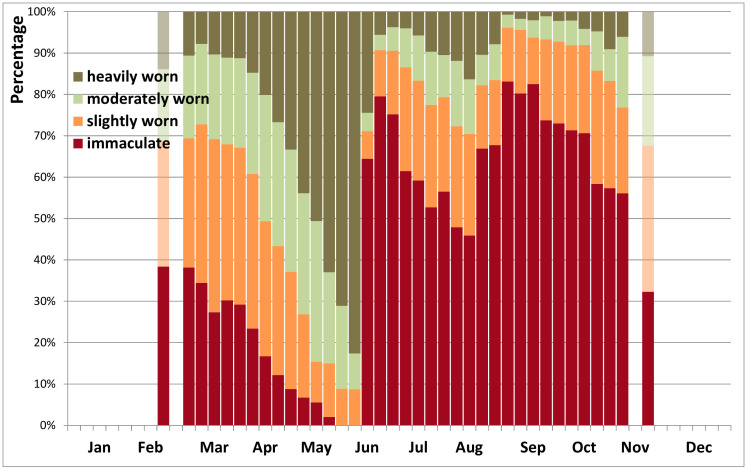
Phenology of wing wear of peacock butterflies in Belgium 2009–2020: proportional by week (*n* = 12,425). (The lower numbers before week 10 and after week 44 were lumped and plotted at weeks 8 and 46).

**Figure 5 insects-12-00683-f005:**
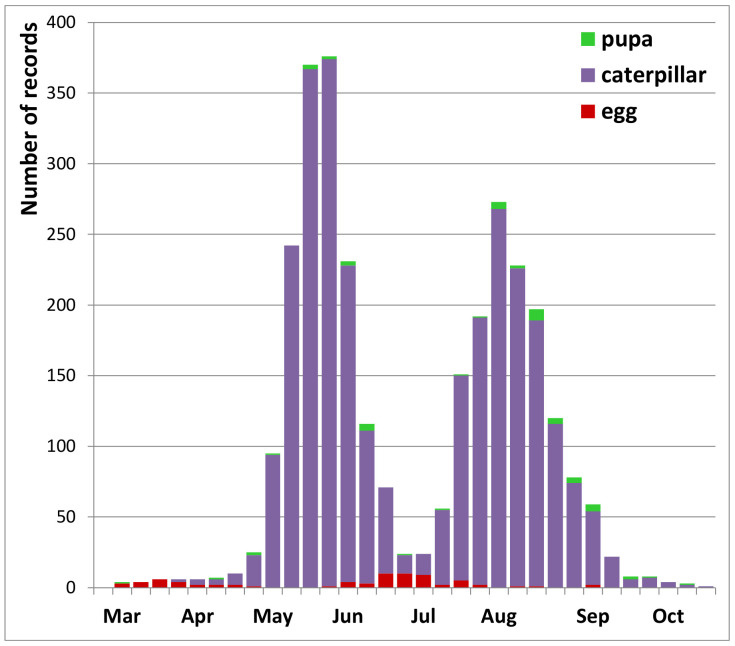
Records of reproduction of peacock butterflies in Belgium 2009–2020 (*n* = 3021 records).

**Figure 6 insects-12-00683-f006:**
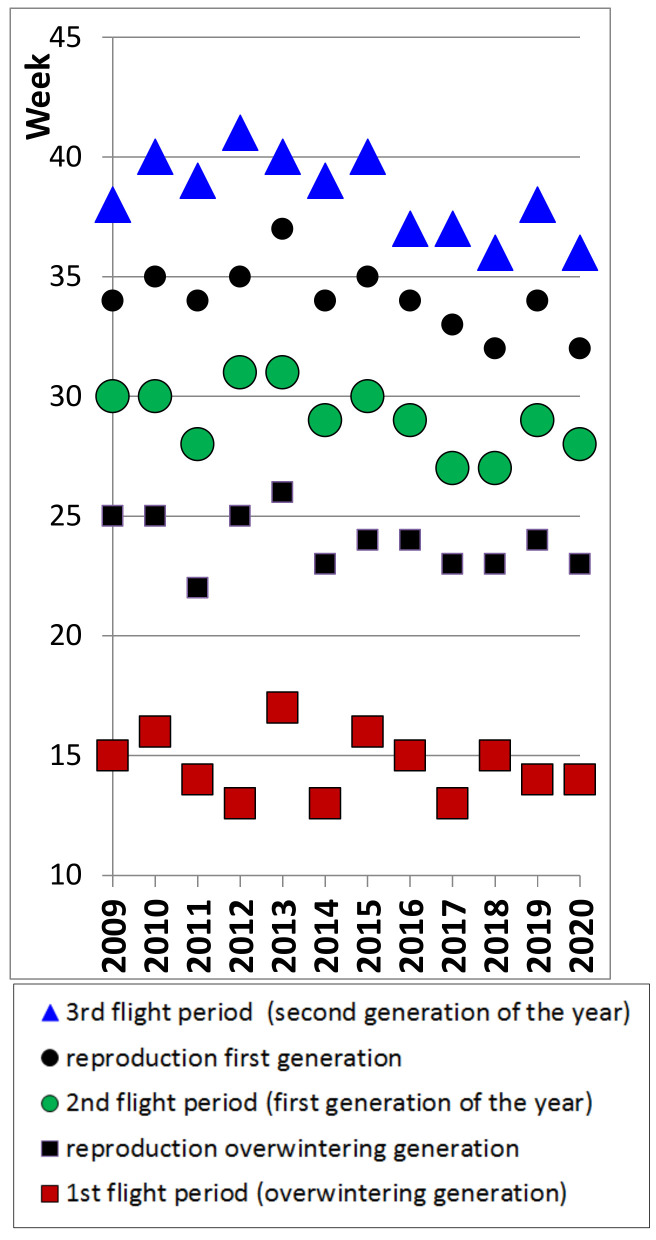
Summary figure with timing (median week number) of flight peaks and reproduction of peacock butterflies in Belgium 2009–2020.

**Figure 7 insects-12-00683-f007:**
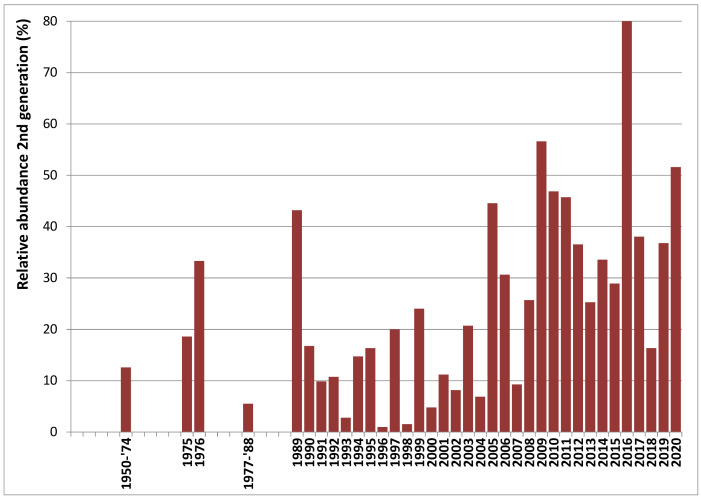
Change in relative abundance of the second generation (third flight peak) of peacock butterflies in Flanders (northern Belgium) since 1950. (1950–2008: 26,933 records; 2009–2020: 319,684 records).

## Data Availability

The full details of all records of peacock butterflies used in this study can publicly be consulted here: https://waarnemingen.be (change website to language of your choice). The aggregated summary data derived from this online platform during this study are openly available in Zenedo repository: (1) Raw data of peacock butterflies at http://doi.org/10.5281/zenodo.5140036; (2) Aggregated data for abundance indices (Figure 2, Figure A8) at http://doi.org/10.5281/zenodo.5139530; (3) Background and summary data for wing wear (Figure 3 and Figure 4) at http://doi.org/10.5281/zenodo.5139947 and http://doi.org/10.5281/zenodo.5138012; (4) Summary data on reproduction (Figure 5) at http://doi.org/10.5281/zenodo.5138224; (5) Summary data on change in voltinism (Figure 7) at http://doi.org/10.5281/zenodo.5138288; (6) Summary data on wear, abundance and photo’s (Appendix B) at http://doi.org/10.5281/zenodo.5139658 and http://doi.org/10.5281/zenodo.5139707; (7) Summary data on abundance of generations (Appendix D) at http://doi.org/10.5281/zenodo.5139532 (links all accessed on 1 February 2021)).

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
