# Peer review of "Abundant Citizen Science Data Reveal That the Peacock Butterfly Aglais io Recently Became Bivoltine in Belgium"

_insects, 2021, doi:10.3390/insects12080683_

Round 1

Reviewer 1 Report

This is an interesting paper, widely based on the involvement of citizens; this was possible because the species target is very easily identifiable.

Author Response

Responses to Reviewer 1 comments

Point 1 (lines 227-231):   This means that the species undergoes some fluctuations with subsequent abundant and scarce years.

Response 1: Are we right to consider this more as a comment than as a call to action? Nevertheless, we added the following sentence: The second generation therefore contributed most to the strongest annual fluctuations.

----------------------------

Point 2 (line 254):   Can you try to correlate it (Figure 7) with mean spring temperatures ?

Response 2: Response 2:  Indeed, one would expect a straightforward relationship here with spring temperatures. But it is more complicated: spring temperatures are only part of the story (as they cause advancement of phenology), but early summer temperatures are as important (they cause more rapid development). Furthermore, the relation is not linear, because once over a certain temperature sum, drought risks are rapidly increasing, resulting in butterflies that intend to (re)produce a second generation, but this turns eventually out to be poor, because many caterpillars die. We have worked through all these relations and have a pretty good understanding of how it exactly works, but that is another story which takes several thousand more words (and many figures). We prefer to give this the full detail it merits in a second paper.

----------------------------

Point 3 (line 298):   Is there an explanation about the decline of this species (Aglais urticae) ?

Response 3:  We have added the following text and references: “… the decline is probably related to climate warming [38,39,73], … “.

[73] Gripenberg, S.; Hamer, N.; Brereton, T.; Roy, D.B?; Lewis, O.T. A novel parasitoid and a declining butterfly: cause or coincidence? Ecological Entomology 2011, 36, 271–281. https://doi.org/10.1111/j.1365-2311.2011.01269.x

Reviewer 2 Report

This is highly valuable paper. using extensive citizen science / butterfly recording data to infer voltinism patterns in one of the most common and popular European butterfly - the peacock Inachis io - which demonstrabyl increased the number of annual generations in the study region, Belgium, during last few decades. This nymphalid species, with overwintering adults, increrased the generation numbers by "inserting" a short non-diapausing summer generation between spring and autumn adult emergence peaks. As following generation numbers in widespread and highly mobile insects is rather difficult - it cannot be done locally, in strictly bound local populations - the authors did tremendous job in amassing evidence from wear of photographed individuals, findings of immature stages, records of reproductive behaviour, etc. etc. Their impressive synthetic study thus also illustrate how laborious it can be to answer seemingly simple questions on common species. 

I have two minor remarks. One concerns detection of the pattern in Belgium, which is middle-sized (by European standards, so small by global standards) country with prevailingly flat relief (ok, there are the Ardennes). Further eastwardly, in Germany, Czech Republic, etc., voltininsm patterns in common species may by obscured by elevation differences, species being bi- or univoltine depending on altitude, and by individual(!) lifetime movements across altitudes. I believe that appropriate reference must exist for these obscuring situations, and a note about it should appear somewhere in lines 298-301. This may partly explain "collapses" of species such as A. urticae and A. levana in low-elevated but not mountainous areas. 

Second, regarding the use of wing wear to infer generation numbers, a similar approach was recently used by Vodickova et al., Journal for Nature Conservation 2019, 52, article 125755, to infer voltinism increase in the checkerspot Melitaea didyma. Mentioning this would be just nice. 

Author Response

Responses to Reviewer 2 comments

Point 1 (lines 298-301):   Further eastwardly, in Germany, Czech Republic, etc., voltininsm patterns in common species may by obscured by elevation differences, species being bi- or univoltine depending on altitude, and by individual(!) lifetime movements across altitudes. I believe that appropriate reference must exist for these obscuring situations, and a note about it should appear somewhere in lines 298-301.

Response 1: We agree that altitudinal movements are an important aspect (from which we were spared), and have inserted the following comment to line 278 of the discussion: We had the advantage in this study that Belgium is a relatively flat country whereby the local phenological patterns are little at risk of becoming obscured by individual butterflies undertaking altitudinal movements from zones with a different phenology or voltinism.

----------------------------

Point 2:   Second, regarding the use of wing wear to infer generation numbers, a similar approach was recently used by Vodickova et al., Journal for Nature Conservation 2019, 52, article 125755, to infer voltinism increase in the checkerspot Melitaea didyma. Mentioning this would be just nice.

Response 2: A valuable addition. We have added the reference in line 120 of the methods: Like Vodičková and colleagues [67], we derived voltinism from wing wear.  

[67] Vodičková, V.;  Vrba, P.; Grill, S.; Bartonova, A.; Kollross, J.; Potocký, P.; Konvička, M. Will refaunation by feral horse affect five checkerspot butterfly species (Melitaea Fabricius, 1807) coexisting at xeric grasslands of Podyji National Park, Czech Republic? J. Nat. Conserv. 2019, 52, 125755. https://doi.org/10.1016/j.jnc.2019.125755.

Reviewer 3 Report

This is a well-written manuscript with a good research design. They examined sufficient numbers of citizen science records and presented results visually and conclusively. I have only minor comments.

Figure 1: Is it possible to distinguish “immaculate” and slightly worn” conditions reproducibly? As authors mentioned, four categories are not equally weighed, and authors’ justification for this categorization is reasonable. However, categorizing field pictures may be difficult, because of different photography conditions (including hardware and climate). Indeed, I could not distinguish these two photos (immaculate vs slightly worn) shown in Figure 1. Maybe, additional sample photos for these conditions in Figure 1 may help. Arrows for where to watch in the slightly worn wings (if possible) may also help.

Figure 7: This is a nice figure, but my question is: Is the relative abundance of 2nd generation (%) correlated with temperature? Correlation analysis may be performed here.

Section 3.5: In the first paragraph, I found “When the abundance of the second generation reaches 15-25% of the annual total, there is partial bivoltinism” and “Once the third flight peak reaches over 25% of the annual total, the majority is bivoltine”. But there are no explanations to justify these statements. Please explain.

Throughout the manuscript, figures are too large for me. They can be made smaller.

I believe that Appendix and Supplemental Materials are two different sections, but this manuscript says, “Appendix A: Supplemental material 1”. Remove “Supplemental material 1”.

Appendix figures should be labeled as Figure A1, A2, and so on. In the current manuscript, figures in Appendix A are labeled as Figure 1, 2, and so on.

Figures A3, A5, and A6: Please use period for decimal point but not comma.

Figure B1-B12: These figures may be combined into a single figure if each panel is sufficiently small, which is acceptable as an Appendix figure.

Author Response

Responses to Reviewer 3 comments

Point 1:   Figure 1: Is it possible to distinguish “immaculate” and slightly worn” conditions reproducibly? As authors mentioned, four categories are not equally weighed, and authors’ justification for this categorization is reasonable. However, categorizing field pictures may be difficult, because of different photography conditions (including hardware and climate). Indeed, I could not distinguish these two photos (immaculate vs slightly worn) shown in Figure 1. Maybe, additional sample photos for these conditions in Figure 1 may help. Arrows for where to watch in the slightly worn wings (if possible) may also help.

Response 1: You are correct: as always when splitting a continuous gradient into categories, there are border cases that will be judged differently by different operators. However, we ran tests of classification with different persons, and although cases shifting one category up or down occurred, average values did not differ significantly. We also rejected a large amount of photo’s that were not good enough to be judged safely. We have added in appendix 1 enlarged figures 1a&b with arrows indicating the points of wear, and we have also added in appendix 1 references to a further series of reference pictures.

----------------------------

Point 2:   Figure 7: This is a nice figure, but my question is: Is the relative abundance of 2nd generation (%) correlated with temperature? Correlation analysis may be performed here.

Response 2:  Indeed, one would expect a straightforward relationship here with spring temperatures. But it is more complicated: spring temperatures are only part of the story (as they cause advancement of phenology), but early summer temperatures are as important (they cause more rapid development). Furthermore, the relation is not linear, because once over a certain temperature sum, drought risks are rapidly increasing, resulting in butterflies that intend to (re)produce a second generation, but this turns eventually out to be poor, because many caterpillars die. We have worked through all these relations and have a pretty good understanding of how it exactly works, but that is another story which takes several thousand more words (and many figures). We prefer to give this the full detail it merits in a second paper.

----------------------------

Point 3: Section 3.5: In the first paragraph, I found “When the abundance of the second generation reaches 15-25% of the annual total, there is partial bivoltinism” and “Once the third flight peak reaches over 25% of the annual total, the majority is bivoltine”. But there are no explanations to justify these statements. Please explain.

Response 3:  Indeed, these categories of partial and fully bivoltine are arbitrary, however, not without reason. If a species has three flight periods in a year, every flight period is then expected to take some 30% of the total. Anything substantially below that indicates that the third generation was only partial (or failed).  Anything below 15% indicates that there were only limited attempts at bivoltinism. We changed an adapted the text of lines 244-248 accordingly:  With three flight peaks, the third flight peak (second generation) can be expected to take up some 30% of the annual total. When the abundance of the second generation reaches only 15-25% of the annual total, there is rather partial bivoltinism, with part of the adults of the first generation reproducing the same year and others going in dormancy without reproducing, or the second generation failed (like in 2018). Once the third flight peak reaches over 25% of the annual total, we can assume that the majority is bivoltine.

----------------------------

Point 4: Throughout the manuscript, figures are too large for me. They can be made smaller.

Response 4:  I presume this point can be taken up by the lay-outer? If figures are reduced, I hope lettering is sufficiently large. Please contact me in case that would be a problem.

----------------------------

Point 5: I believe that Appendix and Supplemental Materials are two different sections, but this manuscript says, “Appendix A: Supplemental material 1”. Remove “Supplemental material 1”.

Response 5:   I have renamed the additional material, which is now in appendices A-D.

----------------------------

Point 6: Appendix figures should be labeled as Figure A1, A2, and so on. In the current manuscript, figures in Appendix A are labeled as Figure 1, 2, and so on.

Response 6:   I have renamed the figures in appendices appropriately.

----------------------------

Point 7:  Figures A3, A5, and A6: Please use period for decimal point but not comma.

Response 7:   Figures are replaced by new figures with decimal point.

----------------------------

Point 8:  Figure B1-B12: These figures may be combined into a single figure if each panel is sufficiently small, which is acceptable as an Appendix figure.

Response 8:   Not easy to squeeze 12 Figures onto one page, but attempt is inserted in the revised MS. Lay-out can probably do better than me.

----------------------------

Round 2

Reviewer 1 Report

Now the paper may be published.

Reviewer 3 Report

I have no point for further revision.